# Distal Aneurysms of Cerebellar Arteries—Case Series

**DOI:** 10.3390/brainsci10080538

**Published:** 2020-08-10

**Authors:** David Krahulik, Miroslav Vaverka, Lumir Hrabálek, Štefan Trnka, Martin Kocher, Marie Cerna

**Affiliations:** 1Neurosurgery Clinic, University Hospital Olomouc, Nová Ulice, 779 00 Olomouc, Czech Republic; miroslav.vaverka@fnol.cz (M.V.); lumir.hrabalek@fnol.cz (L.H.); stefan.trnka@fnol.cz (Š.T.); 2Radiology Clinic, University Hospital Olomouc, Nová Ulice, 779 00 Olomouc, Czech Republic; martin.kocher@fnol.cz (M.K.); marie.cerna@fnol.cz (M.C.)

**Keywords:** aneurysm, distal cerebellar distal arteries, arteriovenous malformation, clipping, endovascular treatment

## Abstract

(1) Background: Distal aneurysms of cerebellar arteries are very rare. The authors report their case series of distal aneurysms of the cerebellar arteries solved successfully by microsurgery or by endovascular treatment (Table 1) (2) Materials and Methods: Between January 2010 and March 2020, 346 aneurysms were treated in our institution. Eleven aneurysms in seven patients were located on distal cerebellar arteries and, in three patients, the aneurysms were combined with arteriovenous malformations. There were four women and three men, ranging from 50 to 72 years of age. Five patients presented with different grades of subarachnoid hemorrhage or intraventricular bleeding, and two patients were diagnosed because of headache. Aneurysm location was the posterior inferior cerebellar artery in six cases, the superior cerebellar artery in three cases, and the anterior inferior cerebellar artery in 2 cases. One patient had three aneurysms, and two patients had two aneurysms. (3) Results: Nine aneurysms were treated by microsurgery trapping or clipping and, in two patients, the associated arteriovenous malformation (AVM) was resected. Two aneurysms were treated by endovascular coiling, and one associated AVM was successfully embolized. Clinical follow-up was a mean of 11.5 months (range, 3–45 months). (4) Conclusion: The authors present their experience with the treatment of 11 peripheral aneurysms on distal branches of the cerebellar circulation in seven patients which were excluded from circulation by microsurgery or endovascular treatment. In three patients, the associated AVM was treated (two with microsurgery, one with embolization).

## 1. Introduction

Aneurysms arising from the superior cerebellar artery (SCA) are very rare, accounting for only 0.3% to 0.7% of all intracranial aneurysms. They occur as a result of both mycotic disease and complex dysplasia along with arteriovenous malformation (AVM), or they may accompany malforming angiitis in systemic diseases; only rarely do they occur alone [1,2,3,4,5,6,7,8,9,10,11,12,13]. Similar limitations in occurrence may be applied to anterior inferior cerebellar artery (AICA) aneurysms, which are mentioned in the literature usually because of their marked symptomatology determined by a close anatomical relationship with the internal auditory meatus and cerebellopontine angle segments [3,4,14,15]. While involvement of the posterior inferior cerebellar artery (PICA) in an aneurysm on the proximal segment is a common feature of aneurysms (up to 6% of all intracranial aneurysms), for involvement of the distal segment, the above-mentioned limitations must be applied. Merely anecdotal cases, limited cohorts, or numeric mentions within large groups dealing with the involvement of posterior cerebral circulation are reported. In the most comprehensive study of aneurysms of the posterior cerebral circulation published so far, three distal aneurysms on the peripheral SCA were reported feeding a malformation, with 13 distal aneurysms on the PICA and a mere four distal aneurysms on the AICA [2,4,6,8,16,17]. The largest study of distal PICA aneurysms (30) was published by Tokimura [18], while Hernandez focused on distal aneurysms of all intracranial arteries with predilection for cerebellar arteries [19]. The incidence of distal aneurysms on cerebellar arteries was 4.3 times greater than that of distal aneurysms on cerebral arteries, indicating a predilection of distal aneurysms for posterior circulation. Distal PICA aneurysms were the most common distal aneurysm on cerebellar arteries. AICA aneurysms had the highest percentage of distal location [19]. The management reflects current trends, with microsurgical clipping, endovascular treatment, or a combination of both; simultaneous revascularization is indicated in exceptional cases.

## 2. Case Description 

### 2.1. SCA Aneurysm

*Anatomy of the SCA:* The SCA belongs to the main arteries supplying the cerebellum. In most cases, it originates from the basilar artery (BA) near the apex and passes below the III nerve (in exceptional cases, it may originate from the posterior cerebral artery/PCA and, thus, pass above the III nerve), continues caudally and goes around the circumference of the brain stem near the pontomesencephalic junction, passes below the IV nerve, and goes on above the V nerve. After passing above the V nerve, it enters the cerebellomesencephalic fissure and then courses posteriorly below the tentorial margin and bifurcates over the cerebellar surface. In 90% of cases, the SCA originates as a solitary trunk which is divided into the rostral and caudal branches during its course. In the remaining 10%, it arises as a duplicate trunk, whereas triplicity is rare.

The SCA is divided into four segments: anterior pontomesencephalic, lateral pontomesencephalic, cerebellomesencephalic, and cortical. Distal aneurysms of the SCA arise on the lateral pontomsenecephalic or distal portions of the SCA.

A 57-year-old female hypertonic patient was admitted to the neurology unit of a regional hospital in a mild Hunt and Hess grade for sudden-onset headache and vomiting. Computed tomography (CT) confirmed Fischer grade 2 subarachnoid hemorrhage (SAH). Following atypical angiography findings, an aneurysm on the bifurcation of the basilar artery (BA) was suspected (Figure 1), and the patient was urgently referred for endovascular treatment. Following catheter placement and 3 Dimensional—Digital Subtraction Angiography imaging, an aneurysm on the peripheral branch of the SCA was found, which was inaccessible for endovascular treatment due to the luminal diameter of the vessel. An additional MRI confirmed the suspected location and gigantic size of the partially thrombosed lesion, and the patient was prepared for surgery. With a midline supracerebellar approach, trapping of the parent artery and extirpation of a large (15 × 18 mm) thrombosed sac with preservation of the precentral vein were performed. Histological findings were without surprise; no mycotic agents or bacterial inflammation in the aneurysmal wall were revealed. Her postoperative course was uneventful, and follow-ups are undertaken every year.

### 2.2. AICA Aneurysm

*Anatomy of the AICA:* The AICA most often originates from the basilar artery as a single vessel (72%), but may also arise as two (duplicate—26%) or three (triplicate—2%) arteries. Then, the AICA encircles the pons near the abducent, facial, and vestibulocochlear nerves. After coursing near and sending branches to the nerves entering the acoustic meatus and to the choroid plexus protruding from the foramen of Luschka, it passes around the flocculus on the middle cerebellar peduncle to supply the lips of the cerebellopontine fissure and the petrosal surface. It commonly bifurcates near the facial–vestibulocochlear nerve complex to form a rostral and caudal trunk. The AICA gives rise to perforating arteries to the brainstem, to choroidal branches to the tela and choroidal plexus, and to the nerve-related arteries. The AICA segments are as follows: 

Anterior pontine segment—between the clivus and the belly of the pons, usually in contact with rootlets of the VI nerve.

Lateral pontine segment—begins at the anterolateral margin of the pons and passes through the cerebellopontine angle above, below, or between the VII and VIII nerves and is intimately related to the internal auditory meatus and the lateral recess. This segment is divided into premeatal, meatal, and postmeatal portions.

Flocculopeduncular segment—begins where the artery passes rostrally or caudally to the flocculus to reach the middle cerebellar peduncle and the cerebellopontine fissure.

Cortical segment—forms terminal branches and predominantly supplies the petrosal surface. Distal aneurysms of the AICA arise on the lateral pontine segment or distally.

A 65-year-old man presented in a peripheral setting with sudden-onset headache. CT revealed SAH at the convexity of the brain and a hemocephalus in the third and fourth ventricles. Angiography (AG) was negative after transfer. A repeat AG 10 days after the SAH showed a saccular aneurysm on a common aberrant segment of the AICA and PICA branches (Figure 2).

MRI confirmed the location of the aneurysm to be at the roof of the fourth ventricle. The patient was prepared for surgery and, with the suboccipital approach, surgical trapping of the parent artery and extirpation of a large partially thrombosed aneurysm were performed. The location of the aneurysm was without difficulties, and the approach was straightforward through caudal vermotomy in the midline to the fourth ventricle. Histological findings were normal. Postoperatively, the patient required temporary drainage of liquid using a lumbar drain, with subsequent surgical repair of a subcutaneous cerebrospinal fluid (CSF) pseudocyst with an excellent long-term result.

### 2.3. SCA, AICA Aneurysms + AVM

A 52-year-old woman presented with atypical headaches on an acute stage. MRI and subsequent PAG revealed an AVM of the cerebellar convexity with three aneurysms on parent arteries at the SCA and AICA bifurcations (Figure 3). The patient was treated by microsurgical clipping and resection of the AVM. Her postoperative course was uneventful, and follow-ups are undertaken every year.

### 2.4. PICA Aneurysms

*Anatomy of the PICA:* The PICA has the most complex, tortuous, and variable course and area of supply of the cerebellar arteries. The PICA generally arises from the vertebral artery (VA) but may also arise from the basilar artery. The site of the origin of the PICA from the vertebral artery varies from below the foramen magnum to the vertebrobasilar junction, with the PICA arising from the posterior or lateral surface of the VA more often than from the anterior or medial surface. Most PICAs bifurcate into a smaller medial and larger lateral trunk. The medial trunk supplies the vermis and the adjacent part of the hemisphere, while the lateral trunk supplies the cortical surface of the tonsils and of the hemisphere. The PICA gives off perforating choroidal and cortical branches, and the cortical arteries are divided into vermian, tonsillar, and hemispheric groups.

The PICA segments are as follows:

Anterior medullary segment—lies anterior to the medulla and extends from the origin of the PICA at the VA to the level of the rostrocaudal line through the most prominent part of the inferior olive that marks the boundary between the anterior and lateral surfaces of the medulla.

Lateral medullary segment—begins where the artery passes the most prominent point of the olive and ends at the level of the origin of the IX, X, and XI nerves.

Tonsillomedullary segment—begins where the PICA passes posterior to the IX, X, and XI nerves and extends medially across the posterior aspect of the medulla near the caudal half of the tonsil. It ends where the artery ascends to the midlevel of the medial surface of the tonsil.

Telovelotonsillar segment—begins at the midportion of the PICA’s ascent along the medial surface of the tonsil toward the roof of the fourth ventricle and ends where it exits the fissures between the vermis, tonsil, and hemisphere to reach the suboccipital surface.

Cortical segment—begins where the trunks and branches leave the groove between the vermis medially and the tonsil and the hemisphere laterally, and includes the terminal cortical branches. Distal aneurysms of the PICA arise from the lateral medullary segment or distally.

A 60-year-old woman was admitted for abrupt loss of consciousness. CT in a peripheral setting showed cerebellar hemorrhage with a pan-intraventricular hematoma in the fourth ventricle and a massive SAH in the cerebellar and spinal canal regions (Figure 4). AG revealed three aneurysms on the distal PICA (Figure 5), successfully excluded from circulation by microsurgery clipping while maintaining patency of the PICA. During the surgery, sudden intraoperative rupture occurred from the most peripheral aneurysm, located just under the arachnoid membrane on the surface of the cerebellum. After easy clipping of the neck, the feeding vessel was followed in the hematoma to its telovelotonsillar part, where another two aneurysms were visible. With the technique of temporary clipping, the optimal final position on both clips was achieved in the wide necked aneurysms. The patency was confirmed with intraoperative ultrasound. A difficult postoperative course and hydrocephalus required temporary ventricular drainage, which was converted to a ventriculo peritoneal (VP) shunt with a good long-term result.

### 2.5. PICA Aneurysms + AVM

A 50-year-old man presented with a headache. Magnetic resonance imaging (MR) and angiography showed cerebellar AVM and two aneurysms on the right side of the PICA (Figure 6 and Figure 7). The patient underwent an operation, and AVM was successfully resected with both aneurysms clipped. The control angiogram showed good result of the treatment (Figure 8), the postoperative course was without complication, and the patient was discharged.

### 2.6. SCA Aneurysms and AVM

A 58-year-old woman with sudden onset of headache and vomiting was admitted to hospital. The CT scan showed subarachnoid hemorrhage (Fischer grade 3), and the CT angiogram revealed AVM with an associated aneurysm on the left superior cerebellar artery (Figure 9). Endovascular treatment was indicated, and successful Onyx embolization of AVM and coil embolization of the aneurysm was performed (Figure 10). The postoperative course was also without complication, and the patient is followed up.

### 2.7. PICA Aneurysm

A 72-year-old women woke up with a severe headache. The CT scan showed SAH (Fischer grade IV). The CT angiogram and digital subtraction angiography revealed an aneurysm of the distal PICA (Figure 11) and, according to the age of the patient and comorbidities, endovascular treatment (Table 1) was indicated. The procedure was without complication with complete occlusion of the aneurysm (Figure 12). Because of the massive SAH, she had ventricular drainage for five days. The control CT scan showed a small ischemic lesion of the left cerebellar hemisphere and an obstructive hydrocephalus that was solved by endoscopic third ventriculostomy. After 12 days in the neurosurgical intensive care unit (ICU), she was discharged to a rehabilitation center with improving neurological status.

## 3. Discussion

In general, aneurysms on the distal branches of the cerebellar arteries are very rare [1,3,4,6,8,10,11,12,13,18,19]; their isolated occurrence is quite exceptional and, more frequently, they accompany mycoses and are associated with arteriovenous malformations or they occur in other generalized diseases of connective tissue.

They are mostly small saccular berry aneurysms; in posterior circulation, however, fusiform aneurysms may frequently occur, and even giant, often partially thrombosed aneurysms are not uncommon [5].

A distinctive feature of the involvement of the periphery of cerebellar arteries is the occurrence of aneurysms on sharply curved segments of arteries without ramification and, not infrequently, an aneurysm can be found on the straight segment of the vessel without an evident location of hemodynamic stress concentration, which is usually considered to be one of the causes of aneurysm formation. Such involvement of the periphery of arteries is not found in anterior circulation. An exception to this constitutes peripheral aneurysms in an infective (most frequently mycotic) condition. In anterior circulation, as well as in the proximal segments of the cerebellar arteries, aneurysms typically involve arteries just after they branch off the main trunk, where even model flow curves show maximum vascular wall stress. A common feature of all distal aneurysms of cerebellar arteries is an extremely high tendency to rupture [9,13,15].

Embryonal development of the posterior circulation is based on the formation of the basilar artery and vertebral arteries from the plexiform formation around the brain stem, where the transverse branches unite with the longitudinal remnant of the lateral canal. A similar plexiform plexus gives rise to the cerebellar arteries. This determines their variability with regard to both the course and the bifurcation, unparalleled in anterior circulation. Embryonal development of combinations of junctions of transverse and longitudinal portions of the plexus could explain the weak spots in the vascular walls and, thus, underlie the development of vascular dysplasia outside the bifurcation region on straight segments.

The coincidence of aneurysm and arteriovenous malformation is as rare as in anterior circulation [2,5,18]. Potentiated hemodynamic stress and associated developmental inferiority of a vessel, as well as the influence of angiogenetic factors upon vessels produced by malformation, are usually considered to be the cause of this coincidence. An aneurysm on the parent vessel may occur either on the proximal or distal segments. The therapeutic strategy after SAH is to firstly exclude the source of hemorrhage from circulation. In microsurgery, once the malformation is excluded from circulation, an increase in pressure in the parent artery occurs, thus increasing the risk of aneurysm rupture. In gradual obliteration of the malformation using endovascular therapy or gamma-knife in some types of proximally located aneurysms, their regression can be anticipated as a result of removal of the hemodynamic stress brought about by subsequent gradual decrease of blood flow due to malformation shunts. 

Large aneurysms are often partially thrombosed. This phenomenon results from the anatomical relations in which the balance of thrombolysis and thrombogenesis is determined by the relatively low blood flow in the vessel and by the ratio between the size of the aneurysm and the size of its neck. Unfortunately, this is an unstable and largely unpredictable condition wherein even a large thrombosed aneurysm may recanalize and bleed repeatedly [10,13].

The anatomical relations of the peripheral arteries of the cerebellum determine some other distinctive features of aneurysms present on them. The AICA and the PICA reach, with their peripheral branches, the walls of the fourth ventricle. An aneurysm in this location can, thus, be freely positioned in the ventricle and ruptures more easily. The hemorrhage then creates a typical pan-intraventricular hematoma in the fourth ventricle along with SAH, with a maximum in the area of the posterior fossa and peri mesencephalic cisterns. Massive subarachnoid hemorrhage in the area of the spinal canal is frequent. An acute hydrocephalus is also frequent here, and a tendency to develop a chronic hydrocephalus is markedly higher than in anterior circulation. This is due to obstruction of CSF pathways by an initial haemocephalus in the fourth ventricle, as well as due to impaired resorption of CSF in the spinal canal. Fenestration of the lamina terminalis with a surgical approach is not possible; therefore, the standard procedure is the placement of a lumbar or, more frequently, ventricular drain. In the prone operative position, the drain can be conveniently placed into the occipital corner of the ventricle. Impaired CSF circulation also leads to frequent problems with wound healing, and liquid drainage reduces the occurrence of liquor fistulae and subcutaneous liquid pseudocysts. Thus, the high proportion of shunt-dependent patients in this type of hemorrhage cannot be affected by endoscopic ventriculostomy of the third ventricle.

AICA aneurysms are characteristic for their relation to the structures of the cerebellopontine angle; thus, they may be mistaken for a typical tumorous lesion of this region. The relation of the aneurysm and an AICA branch, the labyrinth artery, then has an impact on the function of the hearing apparatus. Collateral supply of the labyrinth from the anterior circulation via the temporal bone is nonstandard and cannot be relied on. Therefore, separation of the neural structures when treating an aneurysm is highly delicate, and decompression of the internal auditory meatus is difficult but essential.

Evaluation of the angiogram, CT angiography, and MR angiography raises specific problems when identifying distal aneurysms of the cerebellar arteries. Approximately 20% of initial images are negative. The type of hemorrhage on CT and the combination of the SAH prevalent in the posterior fossa and hematoma in the fourth ventricle must, however, lead to a targeted search for the source of hemorrhage.

Currently, endovascular and microsurgical exclusion of the aneurysm from circulation are competing to become the mainstay of treatment of the complex disability following an SAH. With regard to the luminal diameter of the vessel and the shape of the aneurysm, the method of choice for distal aneurysms of the cerebellar arteries can be microsurgery [1,2,3,4,6,7,8,9,13,14,15,16] or endovascular treatment [17,18,19].

The approaches to SCA aneurysms recommended in the literature are based on the vascular segment involved: subtemporal transtentorial approach for an aneurysm arising from the anterior or lateral pontomesencephalic segment, subtemporal or occipital transtentorial approach for an aneurysm arising from the cerebellomesencephalic segment or from the proximal cortical segment, and infratentorial supracerebellar approach for an aneurysm on the distal cortical branch [3,4,5,8,11,12]. For aneurysms on the periphery of both the AICA and PICA, the retromastoidal approach is sufficient in most cases; however, if necessary, partial condylectomy may be included [5,7,8,9,13,14,15,16]. For aneurysms on the periphery of the arteries in the midline, the medial approach is suitable following suboccipital craniotomy and vermotomy. In atypical locations, neuronavigation is advantageous. Currently, there is no precise indication of revascularization of the periphery of cerebellar arteries [3]. Preoperative testing is not possible, and intraoperative vascular pressure and flow measurement do not yield unequivocal results. Good results can mostly be expected after occluding the peripheral arterial branches in the midline while maintaining an appropriate function of the collateral circulation, and a revascularization, which would be technically very demanding, given the fine caliber of the vessel and the depth of the operational area, is unnecessary. By contrast, with an aneurysm located on the proximal and medial vascular segments anterior to the site of origin of the trunk branches, revascularization with a necessity to occlude is essential for a successful outcome. As donors, the occipital artery or the anterior temporal artery in combination with a venous graft are most often used.

## 4. Conclusions

The authors present their experience with treatment of 11 peripheral aneurysms on distal branches of the cerebellar circulation in seven patients which were excluded from circulation by clipping or microsurgical trapping or by endovascular treatment. According to a review of the literature, microsurgery was preferred to endovascular treatment in this area, but technical improvements in endovascular treatment also make this procedure more effective in the treatment of distal cerebellar aneurysms. Currently, endovascular treatment mostly replaces surgical clipping of the distal aneurysm of cerebellar arteries in elderly patients because of lower morbidity. A well-known tendency to develop chronic hydrocephalus following SAH with the source of hemorrhage in posterior circulation was confirmed in the reported cohort; in two patients with a high Fischer grade, a shunt operation was required, and one endoscopic ventriculostomy was performed.

## Figures and Tables

**Figure 1 brainsci-10-00538-f001:**
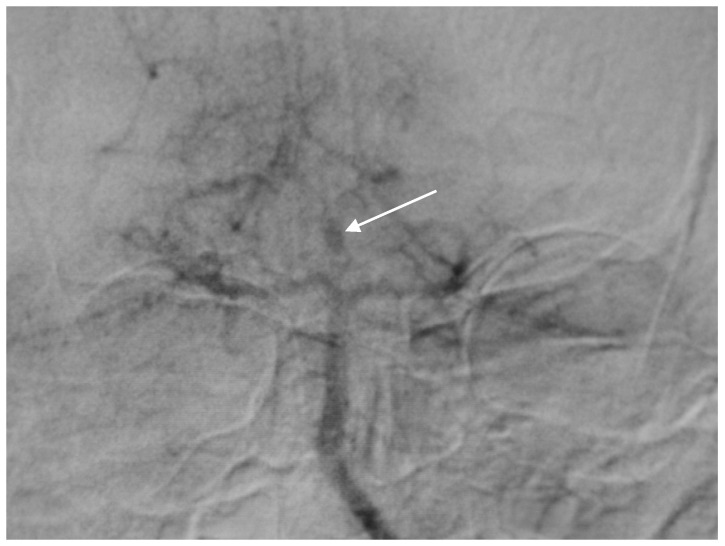
Angiogram misdiagnosed as aneurysm of bifurcation of the basilar artery (BA) in case report 1.

**Figure 2 brainsci-10-00538-f002:**
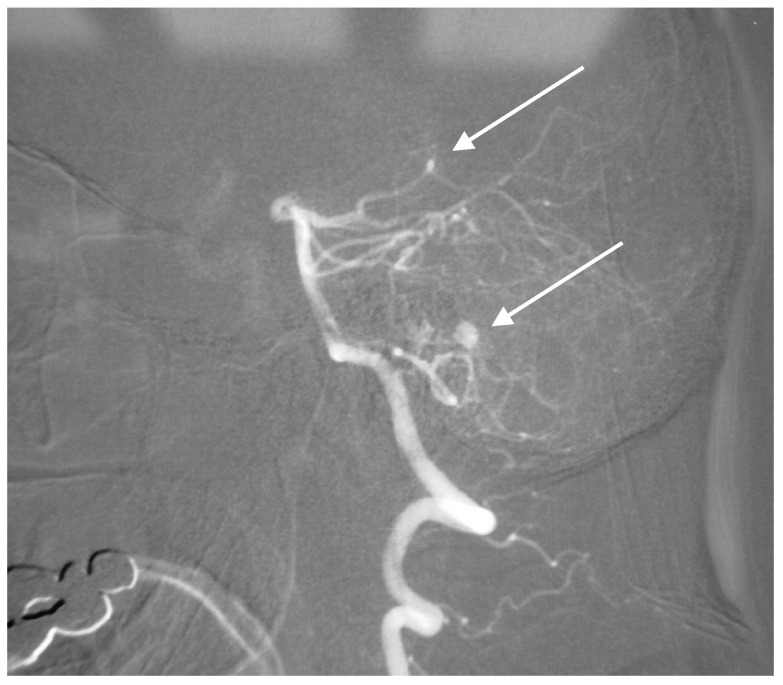
Location of aneurysm on the periphery of a common segment of anterior inferior cerebellar artery (AICA) and posterior inferior cerebellar artery (PICA) on a lateral angiogram.

**Figure 3 brainsci-10-00538-f003:**
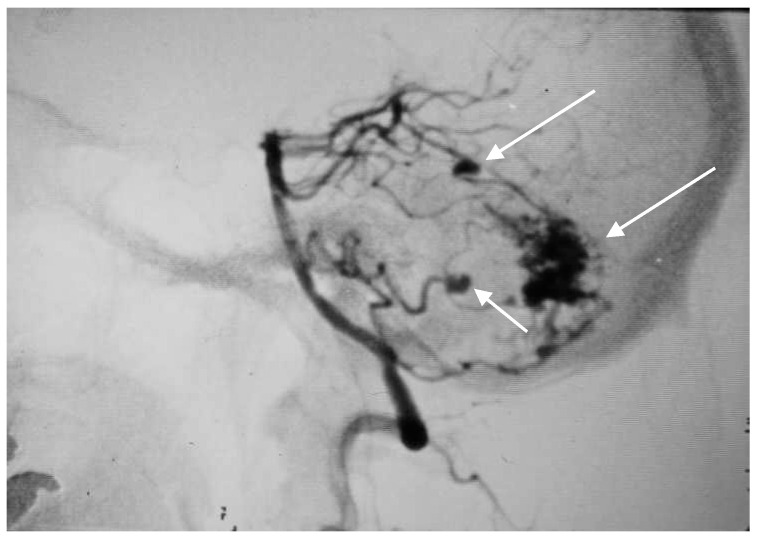
Multiple superior cerebellar artery (SCA) and AICA aneurysms along with cortical arteriovenous malformation (AVM) of the cerebellum on lateral angiogram highlighted with arrows.

**Figure 4 brainsci-10-00538-f004:**
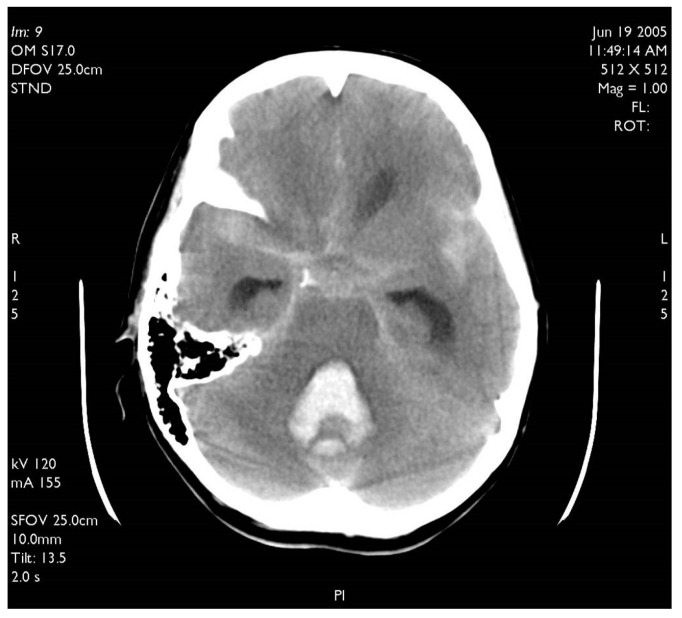
Typical CT scan of subarachnoid hemorrhage (SAH) in posterior fossa with hemocephalus in the fourth ventricle.

**Figure 5 brainsci-10-00538-f005:**
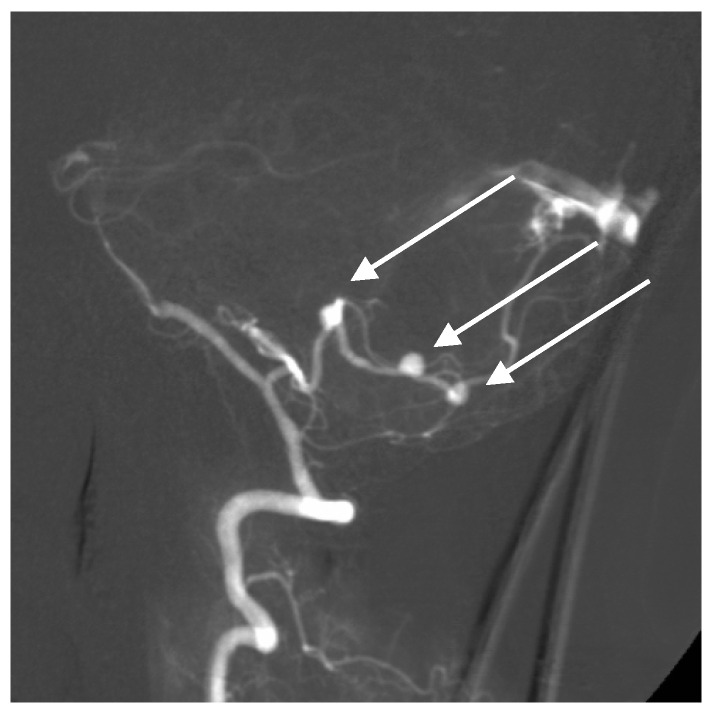
Oblique projection of an angiogram with three PICA aneurysms.

**Figure 6 brainsci-10-00538-f006:**
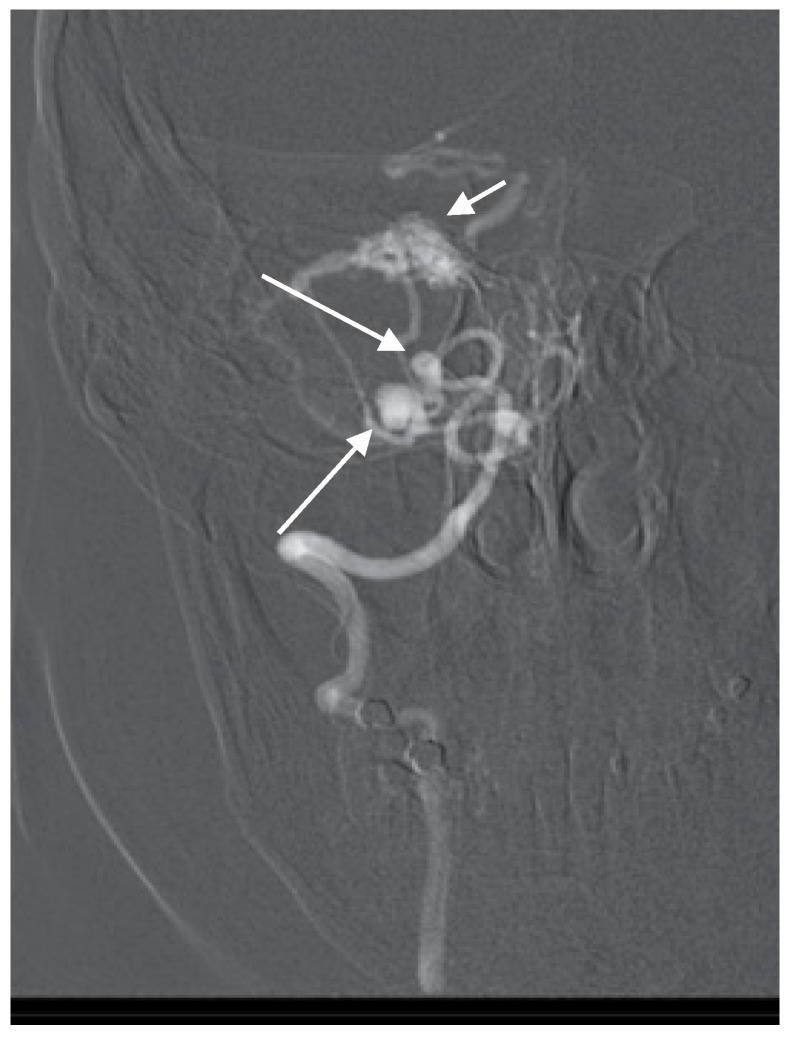
Anteroposterior (AP) angiogram showing cerebellar AVM (short arrow) and two PICA aneurysms (long arrow).

**Figure 7 brainsci-10-00538-f007:**
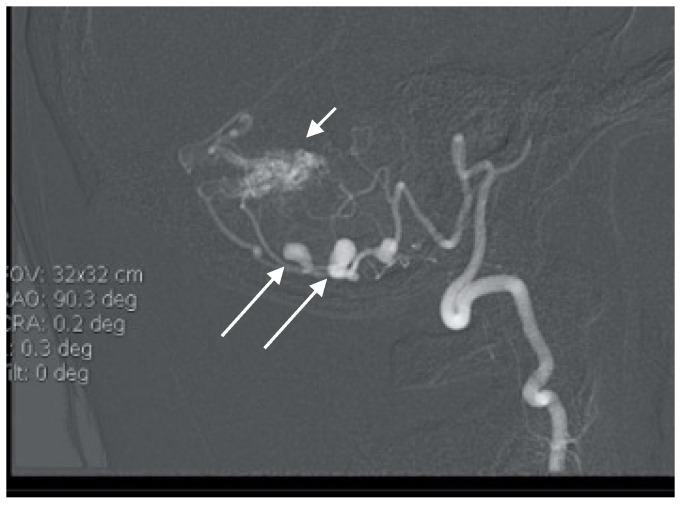
Lateral angiogram showing cerebellar AVM and two PICA aneurysms.

**Figure 8 brainsci-10-00538-f008:**
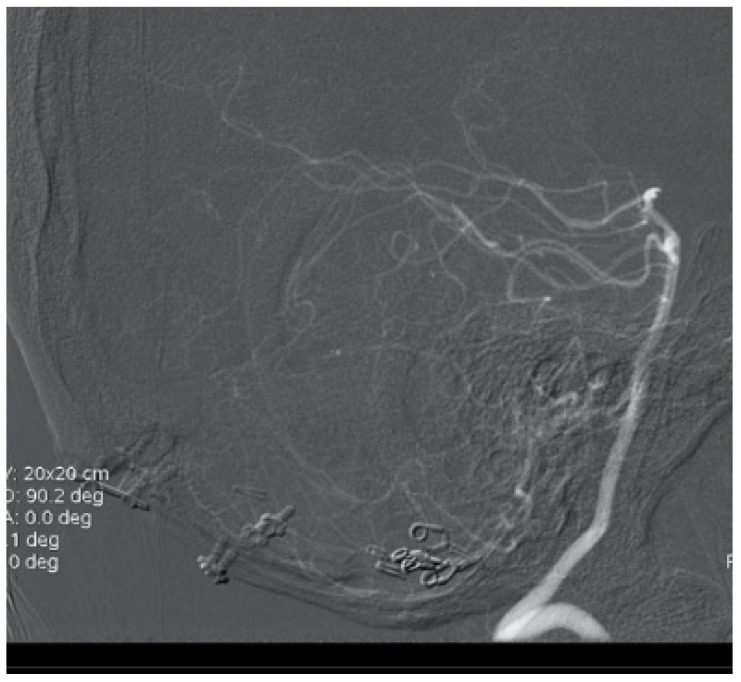
Postoperative control with radical AVM resection and successful clipping of PICA aneurysm.

**Figure 9 brainsci-10-00538-f009:**
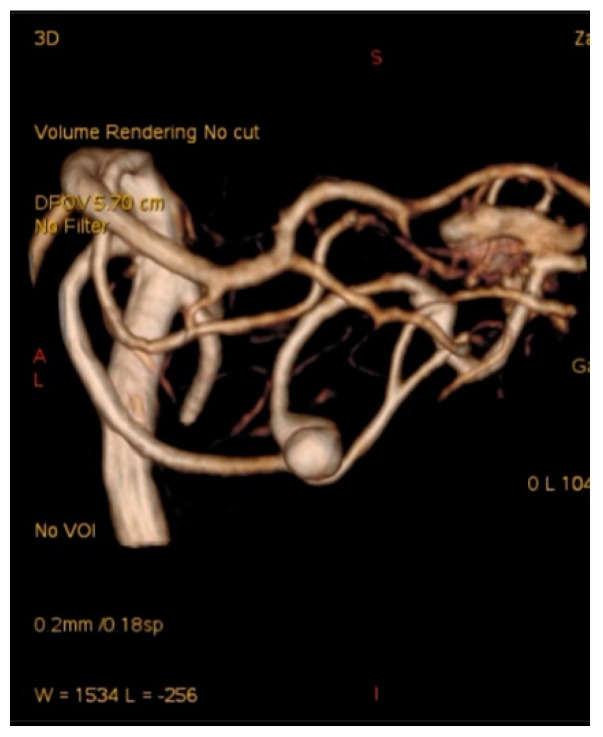
Computer tomography angiography with aneurysm of the left SCA associated with cerebellar AVM.

**Figure 10 brainsci-10-00538-f010:**
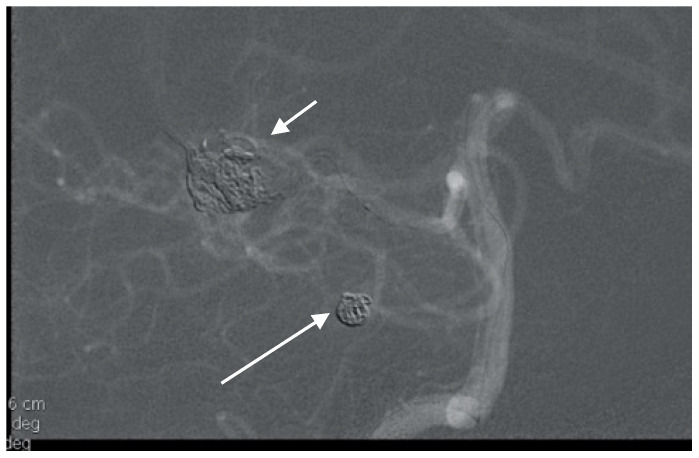
Angiogram showing embolization of the AVM (short arrow) and coiling of the aneurysm (long arrow).

**Figure 11 brainsci-10-00538-f011:**
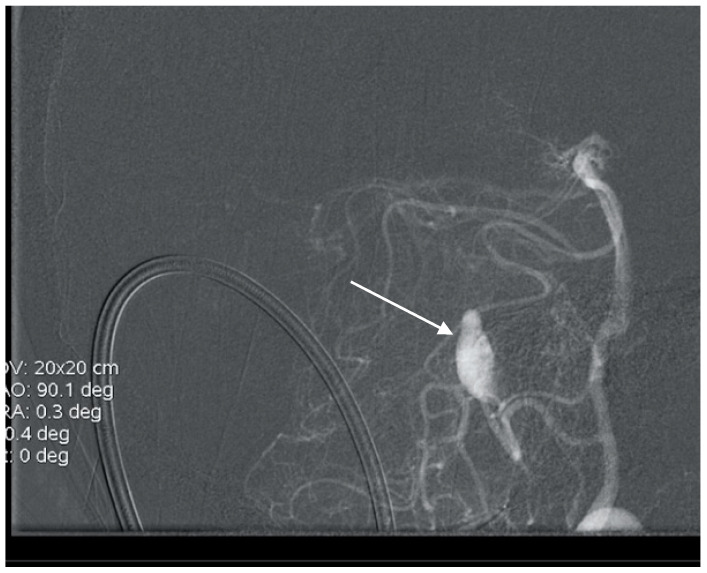
Angiography with distal aneurysm on the left PICA

**Figure 12 brainsci-10-00538-f012:**
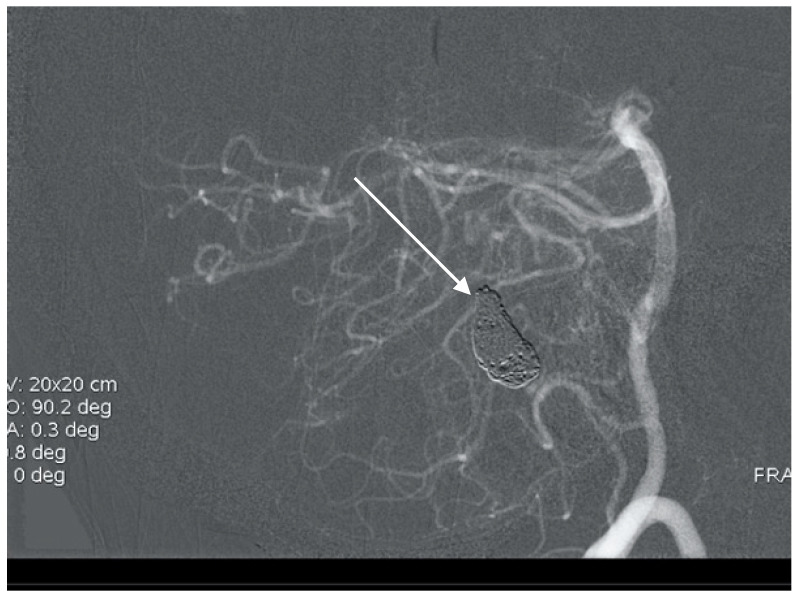
Complete coil embolization of the PICA aneurysm.

**Table 1 brainsci-10-00538-t001:** Clinical data of the patients. F—female; M—male.

Sex	Age	SAH	Aneurysm	Treatment	Associated AVM	Complication
F	57	Yes	SCA	Surgery	No	No
M	65	Yes	AICA, PICA	Surgery	No	CSF pseudocyst
F	52	No	AICA, SCA	Surgery	Yes	No
F	60	Yes	PICA	Surgery	No	VP shunt
M	50	No	PICA	Surgery	Yes	No
F	58	Yes	SCA	Endovascular	Yes	No
F	72	Yes	PICA	Endovascular	No	EVT

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
