# Peer review of "Distal Aneurysms of Cerebellar Arteries—Case Series"

_brainsci, 2020, doi:10.3390/brainsci10080538_

Round 1

Reviewer 1 Report

The authors report about a small series on distal aneurysms of cerebellar arteries. There were 7 patients with a total of 11 aneurysms which were treated by surgery (n=9) or by  endovascular coiling. In three patients the aneurysms were associated with an arteriovenous malformation. The title implies that there is a literature overview on this type of aneurysms which turns out to be very limited and incomplete.

The discussion provides a general overview on various aspects regarding these aneurysms but there are no real new aspects to it.

In summary, although the case series may be of minor interest for a neurosurgical/neurointerventional community, the manuscript does not carry relevant new information neither on the disease nor on the treatment and does not fulfill the expectations of a review of the literature. Therefore I cannot recommend the manuscript for publication in this journal.

The title implies that there is a literature overview on this type of aneurysms which turns out to be very limited. There is quite a number of mansucripts with a significant number of cases which are not cited (e.g. Tokimura et al. Neurosurg Rev (2011) 34:57–67 reporting 30 distal PICA aneurysms or Rodriguez-Hernandez  et al., World Neurosurg. (2013) 80, 1/2:103-112 reporting about 40 distal aneurysms of cerebellar arteries.)

What is the definition of „distal“ aneurysm?

Fig. 7 is missing.

The images could be improved (e.g. center the pathology and exclude wide areas of black fields (e.g. figures 6, 8,11, 12). Figure 1 is rather blurry. Please indicate the aneurysm with an arrow. As it was mistaken as a BA aneurysm it would be nice to have also the additional imaging which confirmed the aneurysm of the SCA.

Abbreviation CTAG in legend of fig. 9 is not explained. There was no DSA performed despite the fact that there was an AVM?

Is there a control DSA after surgery?

In the discussion, surgery and endovascular therapy are regarded as similiar in the treatment of distal cerebellar artery aneurysms. However, to our experience the development of new devices for endovascular treatment has more or less replaced surgical therapy (except when associated with AVM), which is rather invasive and associated with a higher complication rate especially in elderly patients.

Author Response

Thank you very much for your comments. I hope I can answer all of them.

  • The literature was extended with publications focusing at distal aneurysms of cerebellar arteries, that we did not mention in previous text.
  • We added short anatomy of the cerebellar arteries with description of the distal aneurysms.
  • We improved the images with the arrows pointing to the aneurysm and AVM
  • Fig 7 is added
  • 9 – we explained the abbreviation CTAG and the DSA was done with the endovascular treatment in the same time.
  • We added the fact, that in the present time the evolving endovascular treatment mostly replaced the surgical clipping of the distal aneurysm of the cerebellar arteries  especially in elderly patients.

Reviewer 2 Report

The paper is a very interesting articles describing a rare cerebrovascular pathology. Distal aneurysm are uncommon.

The literature includes case reports and retrospective studies with limited cases.

There is little experience on the method to be used to treat this pathology.

The cases presented are not extensive, but considering the frequency, they are sufficiently represented.

The background is clear but in my opinion is not complete. It should be enriched so to provide more information.

Lack a table to summarize the clinical data, (symptoms at presentation), age, gender, type of treatment, complications.

The conclusions must be rewritten by comparing the data the results of the article with those of the literature.

The definition of the acronym PAGA is missing.

The caption of figure 7 is missing

The bibliography must be more precise

Author Response

Thank you very much for your comments. I hope I can answer all of them.

  • We enriched the background and the bibliography of two highly cited papers.
  • We added figure 7
  • PAGA is replaced with Angiography (AG)
  • We added the fact, that in the present time the evolving endovascular treatment mostly replaced the surgical clipping of the distal aneurysm of the cerebellar arteries  especially in elderly patients.

Round 2

Reviewer 1 Report

Thank you for taking up the suggestions and the detailed description of the anatomy. The arrows in the pictures are somewhat off-centered (figg. 1,2,6 and 10) which might be a software problem but should be checked.

Author Response

I have added a short table to summarize the clinical data of the patients and converted the file to the  PDF format  to prevent moving of the arrows on the pictures. 

Reviewer 2 Report

The text of the article has been improved.

I think that could be added a short table to summarize the clinical data of the patients, (symptoms at presentation), age, gender, type of treatment and complications.

Author Response

(The authors gave the same response as above.)
